Lead isotope trends and sources in the atmosphere at the artificial wetland

Cong Ling
Zhai Jiexiu
Yan Guoxin
http://orcid.org/0000-0002-3337-1962 Liu Jiakai
Wu Yanan
Wang Yu
Zhang Zhenming zhenmingzhang@bjfu.edu.cn
Zhang Mingxiang mingxiangzhang2019@163.com
College of Nature Conservation, Beijing Forestry University , Beijing , China
Wang Xinfeng
Electronic publication date: 2019 Oct 16
Publication date: 2019
Volume: 7
Electronic Location ID: e7851
Received 2019 May 28; Accepted 2019 Sep 8
Copyright: © 2019 Cong et al.
Copyright year: 2019
Copyright holder: Cong et al.
License: This is an open access article distributed under the terms of the Creative Commons Attribution License, which permits unrestricted use, distribution, reproduction and adaptation in any medium and for any purpose provided that it is properly attributed. For attribution, the original author(s), title, publication source (PeerJ) and either DOI or URL of the article must be cited.
License URL: https://creativecommons.org/licenses/by/4.0/

Keywords: Total suspended particulate matter, Lead concentration, Lead isotope ratio, EF values, Artificial wetland

Funding: Natural Science Foundation of China 41877535 Fundamental Research Funds for the Central Universities 2016JX05 This research was supported by the Natural Science Foundation of China (41877535) and the Fundamental Research Funds for the Central Universities (2016JX05). The funders had no role in study design, data collection and analysis, decision to publish, or preparation of the manuscript.

==============================
With the rapid development of industry, studies on lead pollution in total suspended particulate matter (TSP) have received extensive attention. This paper analyzed the concentration and pollution sources of lead in the Cuihu Wetland in Beijing during the period of 2016–2017. The results show that the lead contents in TSP in the Cuihu Wetland were approximately equal in summer and spring, greater in winter, and greatest in autumn. The corresponding lead concentrations were 0.052, 0.053, 0.101, and 0.115 ng/m3, respectively. We compared the 206Pb/207Pb data with other materials to further understand the potential sources of atmospheric lead. The mean values of 206Pb/207Pb from spring to winter were 1.082, 1.098, 1.092, and 1.078, respectively. We found that the lead sources may be associated with coal burning, brake and tire wear, and vehicle exhaust emissions. We also calculated the enrichment factor values for the four seasons, and the values were all much greater than 10, indicating that the lead pollution is closely related to human activities.

Introduction

Air pollution, especially particulate matters’ pollution, has become an issue in the public eye in China (Florig, 1997). Particulate matter pollution not only adversely affects human health, but also acts as a catalyst for climate change (Seaton et al., 1995; Kyotani & Iwatsuki, 2002). Studies have shown a positive correlation between air pollution and respiratory system diseases like lung cancer (Dockery et al., 1993). Researchers found that atmospheric aerosols can affect cloud microphysics and indirectly cause changes in light radiation to affect climate (Charlson et al., 1992; Dickerson et al., 1997). Aerosol particles are a mixture of liquid and solid materials which contains trace metals, ions, and organic compounds and so on (Liu et al., 2015). Total suspended particulate matters (TSP) played an important role in analyzing aerosols’ chemical constitution, studying the spatial and temporal variations, revealing the relationship with meteorological factors, and tracing sources (Cong et al., 2018; Ragosta et al., 2002). Atmospheric input of heavy metal elements has a long-term adverse impact on the geobiochemical cycle of ecosystems (Kelly, Thornton & Simpson, 1996). Therefore, it is imperative to understand the heavy metals in TSP.

Trace metals such as Pb, Cd, Hg, and Cr are biologically non-functional and are highly toxic (Salt et al., 1995). Lead has been designated as one of the most dangerous environmental pollutants by the United Nations Environment Programme (Morel, 2008; Shi et al., 2008). With the rapid development of industry, anthropogenic Pb has become the major source of the lead in the environment. They were widespread in the atmosphere, soil, water, plants, and animals (Wang et al., 2013). It is very important to study the geochemical cycle of lead in the environment (Hao et al., 2008; Dawson et al., 2010; Bove et al., 2011; Uzu et al., 2010).

Lead has four stable isotopes which can be used as a tracer of anthropogenic pollution. 206Pb, 207Pb and 208Pb are three radiogenic isotopes while 204Pb is non-radiogenic isotope. These four isotopes can be used as a “footprint” for different sources of lead pollution in the environment, especially for the human activities (Grousset et al., 1994). The inductively coupled plasma-mass spectrometry (ICP-MS) was designed for analyzing stable isotopes more precisely, especially for Pb. The development of ICP-MS made it possible to trace the sources and investigation of heavy metals in different materials. It is widely used to identify the natural sources and anthropogenic pollution (Wiederhold, 2015). The unique lead isotope ratio ranges make it easier to find out the major sources of lead, even though sometimes it may be overlapped (Wang et al., 2013; Bindler et al., 1999; Bollhöfer & Rosman, 2001; Veysseyre et al., 2001; Kaste, Friedland & Stürup, 2003; Zhang et al., 2007). This makes scientists more convenient to identify and quantify the sources of lead in different environmental samples (e.g., atmospheric deposition (Gallon et al., 2005), sediment (Dang et al., 2015), and soil (Huang et al., 2015)), as well as in organisms (Martinez-Haro et al., 2011).

The Cuihu Wetland is the only national urban wetland park in Beijing, which has an area of 1.57 km2. It is one of the most typical artificial wetland, which is constructed to improve the environmental conditions. It also plays important roles in hydrological and economic aspects, especially in keeping biological diversity. However, the Cuihu Wetland is open only in certain days and strictly controls the number of tourists and their activities, which is not like other wetlands in Beijing. Thus, the Cuihu Wetland is less affected by different human activities. This makes the Cuihu Wetland a good place for scientific research. It is reported that artificial wetland is a long term green technology to remove the heavy metals from the polluted areas (Huang et al., 2017), such as Pb. It can also theoretically influence the heavy metal air pollution by increasing humidity and decreasing temperature. The transport of particulates is associated with a series of biogeochemical processes of chemical compounds such as heavy metals (Henderson, 2002). Thus, the variation of heavy metals is also connected with the changes of meteorological factors such as humidity and temperature. This can also provide important information about particle cycling processes (Sun et al., 2016).

However, there were few studies focused on atmospheric lead pollution in an artificial wetland in Beijing. On the other hand, it is difficult to make a systematic research about the lead pollution in the particles by only knowing their total concentrations. Thus, efforts must be made to identify the possible lead sources of the TSPs, thus, to control and reduce the air pollution (Zheng et al., 2004). Therefore, we studied atmospheric lead concentrations and lead isotopic ratios in the Cuihu Wetland in Beijing. We analyzed the temporal variations of lead in TSP in the Cuihu Wetland and compared the differences of lead pollution in atmosphere over different regions and land use types. Another primary target of this study is to determine the sources of lead. We measured lead isotopic ratios in TSP and calculated the enrichment factor (EF) values over a year. Based upon the results, the study attempts to examine the effects of human activity on Pb in the atmosphere and the potential sources of Pb in the TSP in the region. It is helpful for us to have a systematic acknowledgement on the lead pollution in the TSP of the air in an artificial wetland.

Materials and Methods

Sampling site

The Cuihu Wetland is a typical country wetland located north of the Shangzhuang Reservoir in the Haidian District of Beijing. The area of the Cuihu Wetland is 1.57 km2, of which approximately 0.09 km2 is water with an approximate maximum length and width of 1.9 and 1.2 km, respectively.

The weather is rainy and hot in summer (June–September) and dry and cold in winter (December–March). Spring (March–June) and autumn (September–December) are short.

The sampling site was on Crane Island near the center of the Cuihu Wetland (Fig. 1). The island’s main vegetation is willow (Salix babylonica), with reeds (Phragmites communis) growing on the more flat areas of the island.

Figure 1 Position of the sampling site.

Sampling process

An intelligent medium-flow total suspended particle sampler (TH-150; Wuhan Tianhong Instruments Co., Ltd, Hubei, China) and Teflon filters (Beijing RyderCase Instruments Co., Ltd, Beijing, China) were used to collect TSP. A microwave digestion system is used in the key step of the pretreatment. Samples of atmospheric particulates were digested by microwave digestion system, and Al and Pb were determined by ICP-MS. The advantages of microwave digestion system are: quick heating, strong resolution ability and short dissolution time. Besides, the digestion process is in an airtight container. It can save acid reagent and reduce the interference of impurity elements. Its disadvantage is that it needs manual acid driving and it may induce a lower average data. The sampling flow rate was fixed at 100 L min−1. The filters were put in an open plastic bag and conditioned in a constant temperature (25 °C) and humidity (50%) chamber for 24 h before and after sampling (Marcazzan et al., 2001). The filters were transported to and from the sampling site in sealed plastic boxes.

Ambient TSP samples were collected at the sampling site on Crane Island from September 2016 to August 2017. Three samples were collected simultaneously at the site during each of the four seasons during the year. The duration per sample was 12 h (from 08:00 to 20:00).

Chemical analysis

The determinations of lead concentration, aluminum concentration and the lead isotopic composition (206Pb, 207Pb, and 208Pb) were performed via ICP-MS (Bi et al., 2007; Dai et al., 2015). A quarter of a filter sample was first placed in a Teflon digestion vessel. Then, eight mL of nitric acid (6%, v/v) and two mL of hydrogen peroxide were added to the vessel. The vessel was covered and placed in a microwave digestive system to dissolve the sample. The sample digestion was performed then. The first procedure is to heat the samples to 150 °C in 10 min and remaining for 10 min. The second procedure is to heat them to 210 °C in 5 min and remaining for 20 min. Then, the sample solution and filter residue mixture were transferred to a Teflon crucible to heat at 150 °C until nearly dry; five mL of nitric acid (6%, v/v) was then added to the vessel for 15 min to dissolve the filter residue. After cooling, the solution was diluted with nitric acid (1%, v/v) and then used to determine the metal elements. Finally, the solution was measured using an ICP-MS to determine the lead and aluminum concentration and the lead isotopic composition. An international reference material (SRM 981 common Pb isotopic standard) was used for calibration and analytical control before the samples were measured. The precision (% RDS) of the Pb isotopic ratios was typically <0.5%.

Statistical analysis

The statistical treatments of the data were performed using SigmaPlot 12.5 and the IBM SPSS Statistics 22 statistical software.

Enrichment factor analysis

We calculated the EFs to identify the origin of lead and to calculate the proportions of the anthropogenic sources (Ny & Lee, 2010; Yang et al., 2010). In previous studies, these measures have been effective tools to distinguish different sources of heavy metals such as natural sources and anthropogenic sources (Petaloti et al., 2006; Ayrault et al., 2010). The value of EF is calculated via the following relationship: EF=([E][R])sample/([E][R])crust,

where E is the considered element, R represents the reference element for crustal material, ([E]/[R]) sample is the concentration ratio of E to R in the aerosol sample, and ([E]/[R]) crust indicates the mean concentration ratio of E to R in the crust (Han et al., 2006).

Al is abundant in the earth’s crust and is frequently used as a reference element (Han et al., 2006; Taylor & McLennan, 1995; Duan et al., 2012). We calculated the EFs using the value of Al in Chinese soil in 1990, due to the stability and lack of anthropogenic sources. Many of the studies were focused on the concentration changes of different heavy metals. They usually measured the concentrations in surface soils. We found that in 1990, it has been measured of the Al and Pb concentrations of parent rock that Al was 6.62%, and Pb was 26 mg/kg (Wei et al., 1991). If EF approaches unity, the parent rock is the predominant source of the element. Operationally, given the local variation in the soil composition, if EF > 10, it can be assumed that the anthropogenic pollution is the primary source of the elemental abundance (Basha et al., 2010).

Results

Concentration of lead in atmosphere particles

Figures 2 and 3 show TSP and Pb concentrations (±SE) in the samples, respectively.

Figure 2 Seasonal variations in TSP (±SE) expressed in ng/m3 during the study period.

Figure 3 Seasonal variations in the lead concentrations (±SE) expressed in ng/m3 during the study period.

The concentrations of TSP were more than 1,000 times greater than the Pb concentrations. The summer season has the lowest concentrations of TSP at 68.867 ng/m3. The highest concentrations are seen in winter at 244.213 ng/m3. The TSP concentration in spring is higher than that in autumn, with values of 171.528 and 101.042 ng/m3, respectively. However, the seasonal trend is slightly different for TSP and lead. The average concentrations of lead in the four seasons vary from 0.052 to 0.115 ng/m3. The lowest concentration of lead was recorded during summer and spring followed by winter, while the highest concentration was found during autumn at 0.115 ng/m3. The concentrations in spring and summer were 0.052 and 0.053 ng/m3, respectively. The concentration was approximately 0.101 ng/m3 in winter. Even though the concentrations in autumn and winter are higher than those in spring and summer, the only significant difference is between autumn and summer (P < 0.05). There were no significant differences between the other seasons.

Sources of atmospheric lead

The lead isotope compositions in the four seasons are shown in Table 1. In general, the samples show a wide range of lead isotope ratios, ranging from 36.145 to 37.949 for 208Pb/204Pb, from 2.094 to 2.206 for 208Pb/206Pb, from 15.129 to 15.773 for 207Pb/204Pb, from 16.490 to 18.121 for 206Pb/204Pb, and from 1.061 to 1.168 for 206Pb/207Pb (Table 1). 206Pb/207Pb is relatively important in studying the sources of lead in the environment, as it can be determined precisely. The 206Pb/207Pb isotope ratio revealed differences in the behavior in different seasons at the sampling site.

Table 1 Lead isotope compositions.

		Spring	Summer	Autumn	Winter	
208Pb/204Pb	Mean	36.773	37.033	36.795	36.596	
Range	36.710–36.888	36.471–37.949	36.145–37.559	36.233–37.166	
208Pb/206Pb	Mean	2.184	2.162	2.171	2.189	
Range	2.165–2.204	2.094–2.197	2.112–2.206	2.160–2.202	
207Pb/204Pb	Mean	15.568	15.600	15.529	15.517	
Range	15.443–15.688	15.429–15.761	15.129–15.732	15.350–15.773	
206Pb/204Pb	Mean	16.838	17.134	16.957	16.720	
Range	16.678–16.971	16.758–18.121	16.490–17.787	16.505–17.034	
206Pb/207Pb	Mean	1.082	1.098	1.092	1.078	
Range	1.063–1.098	1.069–1.168	1.061–1.132	1.069–1.100	

Enrichment factors

Figure 4 shows the EFs of lead for TSP in the four seasons using Al as the reference element. The EFs represent the enrichment or depletion of lead in the samples.

Figure 4 Lead enrichment factors during the study period.

If an element’s EF value is less than 10, it can be considered to be a crustal (or topsoil) source that is primarily caused by soil- or rock-weathered dust blowing into the atmosphere. If the EF value is much greater than 10, for example, tens to tens of thousands, the element is likely enriched and reflects not just the contribution of crustal material but may also be related to contributions from different human activities.

The EF values in TSP for each season varied substantially from 214 (summer) to 9,623 (autumn). The average EF value of lead is 805 in spring, 557 in summer, 5,133 in autumn, and 3,008 in winter.

Discussion

Variations of lead concentrations in TSP

The Cuihu Wetland is a typical country wetland in Beijing and is little affected by outside conditions in comparison with some industrial sites, which are influenced by heavy metals and related to manufacturing processes. The average lead concentrations in the Cuihu Wetland were low enough, and were even below the safe limits of the international agencies. The WHO and USEPA standard for atmospheric lead is 0.500 ng/m3 (World Health Organization, 2000). During the present study, the average concentration of lead (0.080 ng/m3) was found to be below the limits of the WHO and USEPA standard. The reason for the lower concentration of lead in the atmospheric particulate matter in Cuihu may be self-purification of the wetlands and its distance from large industrial areas. Even though there is a main road which is a practice road for driving school students and a road to transport sand from one sand mining plant. This makes the lead concentrations lower than in areas with many factories or other sources of lead pollution. In addition, the difference between the lead concentrations in the local atmosphere and the WHO level may be due to the different situations of the climate especially the metrological data during the research. We can refine the experimental data by performing additional repetitions and increasing the number of samples.

Variations in average lead levels showed the following sort during the study period: levels in summer were approximately equal to levels in spring, levels in winter were greater, and levels in autumn were the greatest, which is slightly different from a study in Islamabad during the period of 2004–2005, were the levels in summer were approximately equal to the levels in spring, levels in autumn were greater, and levels in winter were greatest (Shah & Shaheen, 2008). The results show that the metal content is inversely proportional to temperature. Even though the concentration of lead in autumn is higher than that in winter, the difference between them is not significant (Kim, Kim & Lee, 1997; Kim, Lee & Jang, 2002; Mishra et al., 2004). Studies have found positive relationships of lead with relative humidity and negative relationships of lead with the temperature (Jonsson et al., 2004; Kim, Kim & Lee, 1997). Other studies show that the wind speed appreciably affects the spread of trace metals. For example, it is shown that the wind speed affects the dilution of lead in the environment (Kim et al., 2002; Vallius et al., 2005; Wu et al., 2002; Ragosta et al., 2002). Furthermore, studies show that the rainfall scavenging is of great efficiency in removing heavy metals from the atmosphere (Mircea, Stefan & Fuzzi, 2000).

Data for lead concentrations in the Cuihu Wetland and other sites are listed in Table 2. We selected nine different types of sampling sites. The lead concentration in the Cuihu Wetland is approximately four to eight times higher than those in wetlands in Taiwan, with values of 0.010 and 0.025 ng/m3, respectively (Fang et al., 2010; Fang & Chang, 2012). The annual concentration of lead in the Cuihu Wetland is similar to that in Haeng Goo Dong, Korea, which was sampled in a grassland (Kim, 2004). Another study of lead in TSP in Beijing had a concentration of 0.690 ng/m3, which exceeds the limit of the WHO and USEPA standard. Okuda et al. (2008) conclude that coal combustion as a major source of some anthropogenic metals. During 1995–2004 there is a large amount of coal for heating supply and residential use in Beijing. Even though that the location for coal combustion for urban residential heating has changed from the domestic stove to large heating supply facilities in recent years. It is also estimated that there is an annual increase in Pb concentration. On the other hand, nonferrous metal smelters are also possible sources. However, efforts must be made to lower the lead concentration. The lead concentrations in forests were very low, followed by grasslands (Wang et al., 2016; Kim, 2004; Quiterio et al., 2006). Lead concentrations appeared higher in industrial areas (Kim et al., 2002; Shaheen, Shah & Jaffar, 2005; Shah & Shaheen, 2007). However, this also depends on the meteorological parameters when the samples were collected and levels are very different in different cities.

Table 2 Lead concentrations in TSP in the Cuihu Wetland and other sites worldwide.

City	Size	Pb (ng/m3)	Season	Character	Reference	
Shenyang, China	TSP	0.115	2013–2014	Farmland	Wang et al. (2016)	
Hailun, China	TSP	0.037	2013–2014	Farmland	Wang et al. (2016)	
Taichung, Taiwan	TSP	0.574	2002	Farmland	Fang et al. (2003)	
Tongyu, China	TSP	0.031	2013–2014	Grassland	Wang et al. (2016)	
Haeng Goo Dong, Korea	TSP	0.084	1991–1995	Grassland	Kim (2004)	
Taejon, Korea	TSP	0.260	2002	Industrial	Kim, Lee & Jang (2002)	
Islamabad	TSP	0.214	2003	Industrial	Shaheen, Shah & Jaffar (2005)	
Islamabad	TSP	0.128	2004	Industrial	Shah & Shaheen (2007)	
Quan-xing, Taiwan	TSP	0.015	2010	Industrial	Fang et al. (2010)	
Chang-hua, Taiwan	TSP	0.019	2010	Downtown	Fang et al. (2010)	
Taiwan	TSP	0.180	2004	Downtown	Wu et al. (2002)	
Beijing, China	TSP	0.690	2005	Residential	Okuda et al. (2008)	
He-mei, Taiwan	TSP	0.016	2010	Residential	Fang et al. (2010)	
Islamabad, Pakistan	TSP	0.144	2004–2005	Urban area	Wang et al. (2016)	
Chang-Hua, Taiwan	TSP	0.034	2009–2010	Urban area	Fang & Chang (2012)	
Changbai Mountain, China	TSP	0.018	2013–2014	Forest	Wang et al. (2016)	
Ilha Grande, Brazil	TSP	0.001	2005	Forest	Quiterio et al. (2006)	
Bei-shi, Taiwan	TSP	0.044	2010	Suburban/Coastal	Fang et al. (2010)	
Gao-Mei, Taiwan	TSP	0.025	2009–2010	Wetland	Fang & Chang (2012)	
Gao-mei, Taiwan	TSP	0.010	2010	Wetland	Fang et al. (2010)	
Beijing, China	TSP	0.080	2016–2017	Wetland	This study	

Sources of lead from nature and human activities

Regardless of the lead sources (lithogenic or anthropogenic), the average 206Pb/207Pb ratio in the four seasons followed the order: summer (1.098) > autumn (1.092) > spring (1.082) > winter (1.078). It is indicated that the geochemical background Pb has relatively high 206Pb/207Pb (approximately 1.200), while low 206Pb/207Pb ratios may indicate potential anthropogenic inputs (Lee et al., 2007). Thus, it could be inferred that in winter one source with low 206Pb/207Pb ratio dominates over others.

The lead isotope compositions of the TSP are helpful to further understand the potential sources of atmospheric lead. We compared the 206Pb/207Pb and 208Pb/206Pb data with that of other materials (Table 3). Due to the Th-rich environment in China, relatively high 208Pb abundances may interfere with estimations of contributions from alkyl lead additives (Chen et al., 2005). Therefore, in the following discussion, we give priority to the influence of 206Pb/207Pb. The results show that the average ratios of TSP in spring are in the range of 1.063–1.098, which is closest to leaded vehicle exhaust (Mukai et al., 1993). In addition, the 206Pb/207Pb isotope ratios in autumn are 1.061–1.132, which are similar to those in spring. The 208Pb/206Pb analysis results are consistent with those of 206Pb/207Pb. Chen et al. (2005) conclude that the 206Pb/207Pb ratios from leaded gasoline range from 1.097 to 1.116. Moreover, Han et al. (2016) indicate that the average 206Pb/207Pb isotope ratio of unleaded vehicle exhaust is 1.147. It seems plausible to assume that the Pb pollution may derive from both the leaded and unleaded gasoline via atmospheric deposition. The Cuihu Wetland is very close to a road, which is a training route for a driving school that has more students in spring and autumn. This indicates that traffic plays an important role in lead emissions. However, unleaded gasoline has been widely promoted in China, which makes high concentrations of lead controversial. Before unleaded gasoline was introduced, lead has been widely released into the environment via leaded gasoline used in vehicles over several decades. Those Pb is of high possibility to be settled into the soil. Such concentrations may be due to the high lead emissions that entered the atmosphere over the past decades, resulting in a relatively high concentration of lead in the soil along the roadside. The movement of vehicles can act to re-suspend dust containing lead into the air (Shah, Shaheen & Jaffar, 2006; Ragosta et al., 2002; Kim, Lee & Jang, 2002; Shah & Shaheen, 2008). It is thought that the sand mining plant near the Cuihu Wetland also plays an important role in increasing the lead concentration. Lead isotope ratios of Chinese coal are reported to vary widely (Mukai et al., 2001). It is interesting that the lead contents in coal are enough low, while they are high in the coal combustion dust samples. This may be due to the fact that combustion process has a “concentration effect” on the emission of lead into the atmosphere (Chen et al., 2005). The 206Pb/207Pb isotope ratios in summer ranged from 1.069 to 1.168, which reflects several factors, such as leaded vehicle exhaust, unleaded vehicle exhaust, and coal. The 208Pb/206Pb ratios indicate that the major source of lead is coal. In winter, the isotope ratios of and 208Pb/206Pb were 1.069–1.200 and 2.160–2.202, respectively. Han et al. (2016) has reported that Pb from coal appeared to be larger than 1.17. It is one of the indicators that one of the sources of lead is coal (Han et al., 2016). Besides, coal is used as winter heating in the Cuihu Wetland. The 206Pb/207Pb values indicate the contribution of three sources: coal, metallurgic dust, and industrial sources. Meanwhile, the 208Pb/206Pb values are very close to those of coal. Increased coal burning in winter is therefore the main source of lead. Trace elements were released into the atmosphere throughout coal combustion via bottom ash, fly ash and gaseous phase. The release of heavy metals depends on the composition of the coal and also on gas temperature and residence time in the flue gas (Mariepierre Pavageau et al., 2002). Studies have also shown that more than 50% of lead in coal may be released into the atmosphere during normal coal pyrolysis processes (Zajusz-Zubek & Konieczyński, 2003). This reminds us that it is of great significance to control the combustion and emission process in ways of reducing the lead pollution in the air.

Table 3 Isotope ratios and the elemental content of possible additional lead sources.

Materials	206Pb/207Pb	208Pb/206Pb	Reference	
Leaded vehicle exhaust	1.11	2.194	Mukai et al. (1993)	
Unleaded automobile exhaust	1.131–1.164	2.106–2.142	Tan et al. (2006)	
Coal	1.153–1.182	2.090–2.220	Mukai et al. (2001)	
Metallurgic dust	1.161–1.185	2.054–2.100	Tan et al. (2006)	
Industrial sources	1.176	2.1	Mukai et al. (2001)	
TSP (Spring)	1.063–1.098	2.165–2.204	This study	
TSP (Summer)	1.069–1.168	2.094–2.197	This study	
TSP (Autumn)	1.061–1.132	2.112–2.206	This study	
TSP (Winter)	1.069–1.200	2.160–2.202	This study	

Enrichment factors of lead

It is obvious that lead in the atmospheric particles came from anthropogenic sources. The highest EF value is found in autumn samples, which also had the highest lead concentration. It can be seen that the EF variation is similar to the trend in the Pb concentrations, which is autumn > winter > spring > summer. These findings indicate that the variation in lead is closely related to human activities. The lead sources are associated with coal burning, brake and tire wear, vehicle exhaust emissions, and the metal industry (Hieu & Lee, 2010; Xu et al., 2013). One possible reason for the high EF values is that the Cuihu Wetland is fairly close to a main road, which is a training route for a driving school. This may increase the opportunity for pollution via brake and tire wear and vehicle exhaust emissions. However, coal burning in autumn and winter also leads to an increase in lead from anthropogenic sources. Other studies have also shown a similar lead enrichment in other places of China (Pan et al., 2015). One study surveyed the EFs in TSP measured at five sites from 2009 to 2010. The results showed that lead was highly enriched in TSP samples in Beijing, Tianjin, Baoding, Tangshan, and Xinglong, with EFs exceeding 100. These high EFs indicate that the lead is of anthropogenic origin, is a key tracer of coal burning (Degen, 1963), and is rich in particles emitted from fossil fuels and biofuel burning (Christian et al., 2009; Wang et al., 2008).

Enrichment factors have been widely used to evaluate the anthropogenic/natural contributions of trace elements (Duce, Hoffman & Zoller, 1975; Polidori et al., 2009). However, the size distribution of the particulate matters or soil samples was another important factor that affects the enrichment of lead (Farao et al., 2014; Li, Wiedinmyer & Hannigan, 2013). Therefore, more efforts must be done to figure out the effect of the size distribution to the sources of lead.

Conclusions

This study showed that the lead concentrations in TSP vary from 0.055 to 0.115 ng/m3 during a year. The average lead concentrations exhibited the following pattern during the study period: the level in summer was approximately equal to that in spring, levels in winter were greater, and levels in autumn were greatest. The lead isotope ratio proved to be a useful tool to characterize the source of the atmospheric lead contamination. Regardless of the lead source, the average 206Pb/207Pb ratio in the four seasons followed the order: summer (1.098) > autumn (1.092) > spring (1.082) > winter (1.078). We also calculated the EF values in TSP for each season. These findings indicate that the variation in lead is closely related to human activities. The sources of lead may be associated with coal burning, brake and tire wear, vehicle exhaust emissions, and the metal industry. We found several possible ways that human activities affect the lead in the environment. However, further effort is needed to decrease and remove such pollution.

Supplemental Information

Supplemental Information 1 TSP concentrations, lead concentrations and isotopes.

Click here for additional data file.

The authors acknowledge the constructive comments provided by both the reviewers and editors.

Additional Information and Declarations

Competing Interests

Author Contributions

Data Availability

The authors declare that they have no competing interests.

Ling Cong conceived and designed the experiments, performed the experiments, analyzed the data, contributed reagents/materials/analysis tools, prepared figures and/or tables, authored or reviewed drafts of the paper, approved the final draft.

Jiexiu Zhai performed the experiments, analyzed the data, contributed reagents/materials/analysis tools, authored or reviewed drafts of the paper, approved the final draft.

Guoxin Yan performed the experiments, contributed reagents/materials/analysis tools, authored or reviewed drafts of the paper, approved the final draft.

Jiakai Liu performed the experiments, analyzed the data, contributed reagents/materials/analysis tools, authored or reviewed drafts of the paper, approved the final draft.

Yanan Wu performed the experiments, contributed reagents/materials/analysis tools, authored or reviewed drafts of the paper, approved the final draft.

Yu Wang performed the experiments, authored or reviewed drafts of the paper, approved the final draft.

Zhenming Zhang conceived and designed the experiments, authored or reviewed drafts of the paper, approved the final draft, help checking grammar.

Mingxiang Zhang conceived and designed the experiments, authored or reviewed drafts of the paper, approved the final draft.

The following information was supplied regarding data availability:

The raw measurements are available in the Supplemental File.

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
