# Peer review of "Lead isotope trends and sources in the atmosphere at the artificial wetland"

_PeerJ, doi:10.7717/peerj.7851_

## Round 0.1 · original submission · Major Revisions

Based on the comments from the two anonymous reviewers, your manuscript needs major revision before considering to be accepted. Detailed information on the sampling site and determination methods should be provided. The basis on the reference element Al and the Pb source of gasoline vehicle require further confirmation or modification according to recently published papers. In addition, some conclusions also need clarification. Please address the major and minor comments and revise your manuscript as soon as possible. Hope you can submit your revision within the next 30 days.

[]

Reviewer 1 ·

Basic reporting

Please, see "General comments for author".

Experimental design

Please, see "General comments for author".

Validity of the findings

Please, see "General comments for author".

Additional comments

The article deals with the seasonal occurrence of lead (Pb) in total suspended particulate mater (TSP) in the Cuihu Wetland in Beijing (China) and the identification of possible Pb sources using stable Pb isotopes, 206Pb, 207Pb, 208Pb and 204Pb.
The main result is that the occurrence of Pb in the atmosphere of the Cuihu Wetland is seasonally variable and the main source of lead is coal burning together with traffic emissions.
I can conclude that the manuscript is original, although not fully novel. On the other hand, this type of research is still interesting, an of high importance, because still a huge number of articles dealt with this topic appear in peer-reviewed literature, mainly in highly impacted journals.
I can imagine that the article may be published in PeerJ, however, some revisions of this manuscript version are urgently needed.
My main comments on the manuscript are the following:
1. I could not find in „Materials and methods“ section no quality control/quality assurrance of the method used for the determination of isotopic Pb composition of TSP samples. It is one of the essential steps to provide precision, accurracy and reliability of analyses performed, especially, in the case of special isotopic analyses.
2. Lines 118 – 132 – You state here that as a reference element, Al was used. However, no information is given in „Materials and methods“ how Al was measured in your TSP samples. It would be also useful to provide the measured concentrations of Al in TSP samples in the main text. Additionally, you write here that as a background value for Al, Al concentration in Chinese soil in 1990 was used for the calculation of EFs, however, it is not clear why Al concentration in Chinese soil in 1990 was used as the background concentration. What was Al concentration? What is Chinese soil in 1990? This is not clear at all. Also, it would be very useful to add what value of Pb background concentration was used. Please, provide better explanation for this and provide it in the text.
3. Lines 209 – 215 – you write here very controversial conclusions. In Lines 155 – 157, a potential reader can read that Pb concentrations in TSP samples are not high because of self-purification ability of the Cuihu Wetland and its distance from pollution sources. However, here, one can read something about high concentrations of Pb and about close proximity of the Cuihu Wetland to some pollution sources, such as road and sand mining plant. Therefore, my question is „What is real situation?“. This must be clarified. Additionally (Lines 219 – 225), you state that different sources of Pb in TSP exist in summer and winter, however, Pb isotopic ratios seem to be not significantly different between summer and winter. Why then do you write about different sources of Pb in TSP of the Cuihu Wetland? Yes, I agree that the coal burning appears to be the main source. Is the coal used as energy source in the close vicinity of the Cuihu Wetland?

Minor comments and notes
- I found different style for reference citation in the main text, chronollogicaly as well as alphabetically. Please, use one style, meet requirements of the journal.
Line 68 – use „...difficult to make a...“
Lines 105-106 – use „Then, 8 mL of nitric acid (6%, v/v) and 2 mL of hydrogen peroxide were added to the vessel.“
Line 118 – „the origin of the heavy elements?; better would be „...the origin of lead...“
Line 122 – „the folume?“. What is it? You mean likely „The value of...“
Line 130 – „the crustal soil?“ Do you think seriously? Please, revise. You mean likely „parent rock“ or „underlying rock“
Line 135 – this should be „Figure 3“ because „Figure 2“ shows TSP concentrations. Please, replace figures in the way that „Figure 2“ will show Pb concentrations in TSP and „Figure 3“ will show TSP concentrations and cite them in the text.
Line 141 and Line 144 – in Line 141 – from 0.55 ng/m3 but in Line 144 you start with lower numbers. Please, revise this carefully.
Line 150 – use „...comparison...“; not „comparson“ and „...related to...“
Line 151 – use „...in the Cuihu Wetland were enough low...“. Please, check the whole manuscript, and use only one form of the wetland name. I prefer using „the Cuihu Wetland“ as in Abstract.
Line 156 – delete „due to the“
Line 166 – use „...is inversely proportional...“
Line 171 – use „For example, it is shown that...“
Line 193 – use „...in studying...“
Line 216 – use „...are reported to vary widely...“
Line 217 – use „...are enough low, while they are high in...“
Line 218 – use „This may be...“
Line 229 – use „Studies have also shown...“
Line 235 – use „...of lead in the samples.“
Lines 261 – use „...important factor...“
Figure 4 – it is not clear what numbers were potted on x-axis. Why Figure 4 is not made as Figures 2 and 3, where x-axis are seasons? Please, explain this or revise Figure 4.

Reviewer 2 ·

Basic reporting

1. English of this manuscript needs as many sentences are unformal. What's more, the format of reference citation requires greatly improvment.

Experimental design

More information on the analytic methods should be added in the text.

Validity of the findings

no commnet

Additional comments

please find in the attachment

Annotated reviews are not available for download in order to protect the identity of reviewers who chose to remain anonymous.

---

## Round 0.2 · Minor Revisions

The revised manuscript has been significantly improved. Based on the first revewer's comments on the revised manuscript, some additional changes are required before it is accepted. The comments are marked by yellow color in the PDF file and require to be addressed. Particularly, the very low value of background Al concentration should be double-checked and carefully adopted with referring to more literatures.

Reviewer 1 ·

Basic reporting

The authors improved significantly their original manuscript.

Experimental design

No comment.

Validity of the findings

The authors used stable isotopic analysis of Pb that was able to characterize the sources of lead in air.

Additional comments

The authors improved significantly their original manuscript. However, despite the improvement of the original version of the manuscript after the first round of review, some additional changes must be done before accepting it.
All my comments and changes that should be done are included directly in pdf-version of the revised manuscript. The majority of the comments is of formal character. My main concern is about the calculation of EF values because the background concentration of Al seems to me to be unreally low, only 6.22 mg/kg!!!. Normal Al concentrations in soils are in a range of thousands up to ten thousands mg/kg.
Please, request the editor for pdf-copy with comments that are highlighted by yellow color.

Annotated reviews are not available for download in order to protect the identity of reviewers who chose to remain anonymous.

Reviewer 2 ·

Basic reporting

The English and referce format of this version have been improved. The figures and tables are neccessary and understandable.

Experimental design

no comment

Validity of the findings

no comment

Additional comments

no comment

---

## Round 0.3 · accepted · Accept

Thank you for your contribution to PeerJ. Hope to see your next work soon.